# Evaluation of the Effects of Heteroaryl Ethylene Molecules in Combination with Antibiotics: A Preliminary Study on Control Strains

**DOI:** 10.3390/antibiotics12081308

**Published:** 2023-08-10

**Authors:** Carmelo Bonomo, Paolo Giuseppe Bonacci, Dalida Angela Bivona, Alessia Mirabile, Dafne Bongiorno, Emanuele Nicitra, Andrea Marino, Carmela Bonaccorso, Giuseppe Consiglio, Cosimo Gianluca Fortuna, Stefania Stefani, Nicolò Musso

**Affiliations:** 1Department of Biomedical and Biotechnological Sciences (BIOMETEC), Università degli Studi di Catania, Via S. Sofia, 89, 95123 Catania, Italy; carmelo.bonomo@phd.unict.it (C.B.); paolo.bonacci@phd.unict.it (P.G.B.); dalida.bivona@phd.unict.it (D.A.B.); alessiamirabile93@gmail.com (A.M.); emanuelenicitra@gmail.com (E.N.); stefanis@unict.it (S.S.); nmusso@unict.it (N.M.); 2Unit of Infectious Diseases, Department of Clinical and Experimental Medicine, ARNAS Garibaldi Hospital, Università degli Studi di Catania, Via Palermo, 95122 Catania, Italy; andreamarino9103@gmail.com; 3Department of Chemical Sciences, Università degli Studi di Catania, Viale Andrea Doria 6, 95125 Catania, Italy; giuseppe.consiglio@unict.it (G.C.); cg.fortuna@unict.it (C.G.F.)

**Keywords:** antimicrobial resistant, heteroaryl ethylene derivatives, QSAR model, antibiotics, antitumor compounds, combination therapy

## Abstract

The discovery of compounds with antibacterial activity is crucial in the ongoing battle against antibiotic resistance. We developed two QSAR models to design six novel heteroaryl drug candidates and assessed their antibacterial properties against nine ATCC strains, including *Enterococcus faecalis*, *Staphylococcus aureus*, *Klebsiella pneumoniae*, *Acinetobacter baumannii*, *Pseudomonas aeruginosa*, and also *Salmonella enterica* and *Escherichia coli*, many of which belong to the ESKAPE group. We combined PB4, a previously tested compound from published studies, with GC-VI-70, a newly discovered compound, with the best cytotoxicity/MIC profile. By testing sub-MIC concentrations of PB4 with five antibiotics (linezolid, gentamycin, ampicillin, erythromycin, rifampin, and imipenem), we evaluated the combination’s efficacy against the ATCC strains. To assess the compounds’ cytotoxicity, we conducted a 24 h and 48 h 3-(4,5-dimethylthiazol-2-yl)-2,5-diphenyltetrazolium bromide (MTT) assay on colorectal adenocarcinoma (CaCo-2) cells. We tested the antibiotics alone and in combination with PB4. Encouragingly, PB4 reduced the MIC values for GC-VI-70 and for the various clinically used antibiotics. However, it is essential to note that all the compounds studied in this research exhibited cytotoxic activity against cells. These findings highlight the potential of using these compounds in combination with antibiotics to enhance their effectiveness at lower concentrations while minimizing cytotoxic effects.

## 1. Introduction

ESKAPE bacteria, including *Enterococcus faecalis*, *Staphylococcus aureus*, *Klebsiella pneumoniae*, *Acinetobacter baumannii*, *Pseudomonas aeruginosa*, *Salmonella enterica*, and *Escherichia coli*, are notorious pathogens known for their ability to evade the effects of multiple antibiotics. The rising tide of antibiotic resistance among ESKAPE bacteria presents a significant challenge to modern healthcare [1], demanding urgent efforts to develop innovative strategies for discovering new antibacterial compounds, fundamental tools to combat this growing threat [2].

Multidrug-resistant bacteria (MDR), comprising methicillin-resistant *Staphylococcus aureus* (MRSA) [3] and carbapenem-resistant Enterobacterales (CRE) [4], are responsible for a significant number of infections worldwide [5]. The emergence and spread of resistance mechanisms among bacterial populations have been fueled by the overuse and misuse of antibiotics [6], horizontal gene transfer [7], and inadequate infection control measures [8]. We recently reported a molecular modeling study and versatile synthetic procedure that allowed us to identify the heteroaryl ethylenes derivatives as a new class of antimicrobial compounds [9]. Among these compounds, PB4 (Figure 1) showed exceptional promise, effectively inhibiting the growth of both Gram-positive and Gram-negative selected ATCC strains and, notably, high-priority clinical strains [10].

In addition to our focus on tackling multidrug resistance, we have also dedicated efforts towards the development of new antineoplastic drugs, encompassing both cytotoxic and adjuvant medicines [11,12,13,14]. Some of the previously synthesized heteroaryl ethylenes have shown remarkable cytotoxicity against the CaCo-2 colon–rectal cancer cell line, leading us to construct an in-house database and conduct a structure–activity relationship (QSAR) study. The structural design approach, when combined with partial least squares (PLS) analyses to predict relevant absorption, distribution, metabolism, and excretion (ADME), as well as biological properties, offers a comprehensive evaluation of in vitro and in vivo efficacy for both antimicrobial and antiproliferative activities [15,16]. Moving forward, the next stage of this ongoing multidisciplinary project involves a combined in silico approach, consisting of three main components: (i) extending the QSAR study to identify novel compounds from our in-house chemical database with potential antimicrobial properties; (ii) developing a new QSAR model to predict cytotoxic activity towards the CaCo-2 colon–rectal cancer cell line; and (iii) selecting promising scaffolds with sufficient potential to advance into a full drug development program. Through multiple in vitro assays, we have assessed the antibacterial activity and toxicity properties of these compounds, demonstrating their effectiveness against various ATCC bacterial strains. Subsequently, we tested these compounds individually and in combination with commonly used antibiotics in clinical practice (such as linezolid, gentamicin, ampicillin, erythromycin, rifampin, and imipenem) on CaCo-2 cells to evaluate their cytotoxicity. This study represents the third installment in a series exploring heteroaryl ethylene compounds with antibacterial activity, and its findings could hold significant implications for the future of antibiotic development and the fight against MDR bacterial infections [17,18].

## 2. Results

### 2.1. Structural Design of Heteroaryl Ethylenes

It is well-established that inflammation arising from chronic bacterial infections or abnormal gastrointestinal colonization is closely associated with colorectal cancer [19,20,21]. Gaining a deeper understanding of these interactions will unlock future opportunities for preventing and treating both bacterial infections and colorectal cancer.

Heteroaryl ethylene compounds occupy a relative niche position in the fields of electronic materials thanks to their electrochemical and optoelectronic properties. Although their precise mechanism of action remains unknown, our recent studies are aiming to broaden their scope of application towards biomedical uses [9,10,14]. This prompted us to perform a systematic investigation of the relationships between molecular structures and their biological activity. We extended our multidisciplinary study starting from an in silico selection of new heteroaryl ethylenes as antimicrobial and antitumor agents.

Predicting the biological properties of the heteroaryl ethylene derivatives is not a standard procedure due to a limited number of research results, and thus, our QSAR models were developed using a small set of data from the literature for the inhibitors of *S. aureus* and an in-house dataset of compounds for the antitumor activity against the colon–rectal cancer cell line CaCo-2. The new drug candidates (see Figure 2) have been selected to retain some structural figures of the lead compound PB4, namely the double branched D-π-A-π-D motif, through the insertion of different donor units (GC-VI-10, GC-VI-70, GC-VII-39, and BCM6), or the 4-(dimethylamino) styryl unit through the variation of the acceptor moiety (GC-VII-50 and BCM4). Therefore, the twofold objective of this preliminary screening was to analyze the potential of these structures as drug candidates for treating both bacterial infections and colorectal cancer, as well as to identify common structural features that correlate with their biological activity.

#### 2.1.1. QSAR Model for the Screening of the In Vitro Activity against *S. aureus* ATCC29213

The data on the antimicrobial activity, expressed as MIC (mg/L), of 53 heteroaromatic compounds tested against *S. aureus* ATCC29213 define our library for predicting antibacterial activity. This dataset comprises the 40 compounds used in our previous QSAR study, along with the 8 heteroaryl ethylenes tested in that same study [9], 3 compounds from our in-house database that were previously tested against *S. aureus* ATCC29213, and some recently reported styryl derivatives [22].

Overall, the PLS model has been enhanced by incorporating 13 new compounds, primarily heteroaryl ethylenes molecules with various donor–bridge–acceptor (D-π-A) motifs (refer to Appendix A). The PLS 3D score plot (Figure 1) illustrates that the most active compounds are positioned on the left-hand side of the plot with negative LV1 scores, and overall, the model exhibits a strong correlation between the experimental and predicted activity (R^2^ = 0.88). The leave-one-out (LOO) procedure allowed for internal validation of the model, resulting in a significant improvement in its predictive ability, with a Q^2^ value of 0.59 (see Appendix A for R^2^ vs. Q^2^ plot).

This molecular modeling approach provides valuable information about the key molecular properties, such as molecular interaction field (MIF)-based descriptors, which correlate with the biological activity (see variable influence on projection VIP and weights plot in Appendix A). Among the 128 VS+ descriptors, the bacteriostatic activity against *S. aureus* ATCC 29213 was significantly influenced by certain factors, including the percentage of unionized species at higher pH (%FU), 3D pharmacophoric descriptors for H-bond donor regions (DODODO, ACDODO, and DRDODO), H-bond donors volume (WO), physicochemical properties such as the distance of the hydrophobic volume from the center of mass (ID), the partition coefficient water/cyclohexane (LogPc-Hex), as well as ADME properties, such as skin and CaCo-2 cell permeability (SKIN and CaCo-2). Additionally, the improved PLS model highlights the correlation of the activity with the compound’s solubilities computed at various pH values and the corresponding shape of the solubility profile curve (LnLgS and LgS).To identify a new promising scaffold, we used this model for the external prediction of the bacteriostatic activity of six new structures (Figure 2) from our in-house database of heteroaryl ethylenes (the yellow circles in Figure 1 indicate the predicted compounds). Compound GC-VI-10 was located beyond a 95% degree of confidence of the scores plot, while the remaining candidates were located within a 99% degree of confidence. Moreover, the most significant aspect is their projections in the region of active antimicrobial compounds active against *S. aureus* ATCC 29213.

#### 2.1.2. QSAR Model for Cytotoxic Activity towards CaCo-2 Colon–Rectal Cancer Cell Line

For the development of the QSAR model to evaluate the cytotoxic activity, we resorted to an in-house dataset of compounds previously synthesized and tested by our groups for their potential antitumor activity against the CaCo-2 colon–rectal cancer cell line. Based on these results, we created a database composed of 38 heteroaryl ethylene compounds and their antiproliferative effect against CaCo-2 (Appendix A).

A preliminary principal component analysis (PCA) applied to the X-matrix of the VolSurf+ descriptors allowed us to gain an overview of the extended dataset. The first three principal components explained 66.9% of the variance, and the PCA score plot is shown in Appendix A. Only three compounds are located at the border of the 95% confidence interval of the plot, while the remaining 35 compounds are located within the 99% confidence interval. No classification of compounds or any training information was given to the PCA model but, interestingly, active and inactive compounds could be nicely discriminated by the first two PCs.

Next, we performed PLS regression using VolSurf+; the PLS 3D score plot (Figure 2) shows that the chemical space is properly explored and provides good discrimination between inactive (blue circles) and active compounds (red circles), with the latter positioned in the lower left side of the 3D plot.

The PLS model for cytotoxic activity towards the CaCo-2 colon–rectal cancer cell line was validated using the leave-one-out (LOO) method, and it presented the following values of statistical coefficients (see Appendix A): R^2^ = 0.86 and Q^2^ = 0.56. The analysis of the weights and VIP plots (Appendix A) revealed that descriptors for the H-bond acceptor regions (WN and DRACACAC), the CaCo-2 cell permeability (CaCo-2), the physicochemical properties such as the distance of the hydrophobic volume from the center of mass (ID), and the hydrophobic volume (CW) were directly related to the activity. On the other hand, descriptors for the ADME properties, such as skin permeability and volume of distribution (SKIN and VD), physicochemical properties used for hydrophobic volumes (CD and DD) or polar surface area (PSAR), and the ratio between the hydrophilic and lipophilic parts of a molecule (CP), were inversely correlated with the cytotoxic activity towards the CaCo-2 colon–rectal cancer cell line.

Since the model seems to clearly separate active from inactive heteroaryl ethylene compounds, we projected the six compounds previously selected into the model for cytotoxic activity prediction. Surprisingly, the results follow the patterns given in the previous PLS model: compound GC-VI-10 was located outside the 95% confidence interval of the score plot, while the remaining candidates are projected in the region of the most active antimicrobial compounds active towards the CaCo-2 colon–rectal cancer cell line.

### 2.2. Impact on Biological Activity and Antimicrobial Susceptibility Test

The compounds selected through the QSAR procedure were tested to determine the minimum inhibitory concentration (MIC). Our results, reported in Table 1, show that heteroaryl ethylenes were active against *S. aureus* (ATCC 29213, ATCC 12598, ATCC BAA-1556), *E. faecalis* (ATCC 29212), *E. coli* (ATCC 25922), and *A. baumannii* (ATCC 17978). The two heteroaryl ethylene molecules that reported the best results were PB4 and GC-VI-70; the former was our ‘lead’ compound [9], and the latter, GC-VI-70, possessed the same double branched D-π-A-π-D structure but two bis-thiophene heteroaryl donor groups.

Surprisingly, BCM4 and GC-VII-39 showed MIC levels ranging from 16 mg/L to ≥128 mg/L for all tested strains. BCM6 was ineffective against Gram-negative bacteria (MIC values from 64 to ≥128 mg/L), whereas it showed lower MIC values for Gram-positive strains (from 4 to 8 mg/L), except for *E. faecalis* ATCC 29212, which reported an MIC value of 32 mg/L. The same trend was also reported for GC-VII-50 (MIC values from 0.25 to 8 mg/L). GC-VI-70 displayed high MIC values for Gram-negative bacteria and better antibacterial activity against Gram-positive strains, with MIC values ranging from 2 up to 16 mg/L. Moreover, all tested strains showed high MIC levels for GC-VI-10, except for E. faecalis ATCC 29212, whose MIC value was 4 mg/L. All tests were conducted in duplicate, and the MIC values are listed in Table 1. Since all replicates show identical values, without variability, the standard error of the mean is zero. Technical positive controls were included to confirm the validity of the procedures according to the EUCAST guidelines for quality control (QC). The QC MIC values are listed in Table 2.

Due to its promising antimicrobial activity and its previously determined cytotoxic profile [9,10], PB4 was selected to undergo broth microdilution (BMD) in combination with commonly used antibiotics, such as linezolid, rifampin, and erythromycin for Gram-positive bacteria, imipenem for Gram-negative, and gentamycin and ampicillin for both, in order to examine its potential to decrease their MIC values.

The data reported in Table 3 show that PB4 led to a notable reduction in rifampin MIC for *S. aureus* USA 300 (4-fold log reduction) and for *E. faecalis* ATCC 29212 (2-fold log reduction). Additionally, the combination with gentamycin led to 2-fold log MIC reductions for *K. pneumoniae* ATCC 700603, *P. aeruginosa* ATCC 27853, and *E. faecalis* ATCC 29212. However, no significant MIC reductions were observed for the combinations with linezolid, erythromycin, ampicillin, or imipenem, for all tested bacterial strains. Specifically, when combined with ampicillin, PB4 demonstrated a two-dilution decrease in MIC for several bacterial strains, including *P. aeruginosa* ATCC 27853, *K. pneumoniae* ATCC 700603, and *E. faecalis* ATCC 29212. In association with rifampin, PB4 decreased the MIC of *E. faecalis* ATCC 29212 by five dilutions (from 0.125 mg/L to 0.006 mg/L); in combination with ampicillin, it decreased MICs by two dilutions for *P. aeruginosa* ATCC 27853 (2 mg/L to 0.5 mg/L), *K. pneumoniae* ATCC 700603 (8 mg/L to 2 mg/L), and *E. faecalis* ATCC 29212 (4 mg/L to 1 mg/L). Furthermore, a positive combined effect between PB4 and the newly tested GC-VI-70 was demonstrated, with a three-dilution decrease in the MIC of PB4 against *E. faecalis* ATCC 29212 (from 0.5 mg/L to 0.06 mg/L) and a two-dilution decrease against *S. aureus* ATCC 12598 (from 0.25 mg/L to 0.06 mg/L). Moreover, since GC-VI-70 displayed the best MIC/cytotoxicity profile (see Section 2.3, with the lowest MIC value (2 mg/L) reported for *E. faecalis* ATCC 29212 and an IC50 value on CaCo-2 cells of 0.32 µM, it was selected to evaluate its potential combined effect with PB4 against Gram-positive bacteria. The combination of the two molecules revealed remarkable efficacy, reducing PB4 MIC values by 2 logs for *E. faecalis* ATCC 29212 and for *S. aureus* ATCC 12598 (see Table 4).

### 2.3. Evaluation of Heteroaryl Ethylene Compound Cell Cytotoxicity

To fully assess the biological activity of the new molecules, they were tested at different concentrations on the CaCo-2 (ATCC HTB-37) colon–rectal cancer cell line. These human cancer cells were exposed to compound solutions ranging in concentration from 100 µM to 0.01 µM. As a reference compound for these assays, we used 5-Fluorouracil (5-FU), a pyrimidine analog from the antimetabolite family. After 24 and 48 h of incubation, we assessed the antiproliferative activity through MTT assays to obtain the cell growth curves and determine the half-maximal inhibitory concentration (IC50) values. The results are presented in Figure 3 and Table 5, respectively.

The results of the MTT assay show that all heteroaryl ethylene compounds exhibited significantly higher cytotoxicity than 5-FU [23], which served as the reference antitumor compound in the assay. Among the compounds tested, BCM6, GC-VII-39, and GC-VII-50 displayed the highest cytotoxicity at 24 h, with IC50 of 0.16, 0.23, and 0.17 μM, respectively. In comparison, 5-FU showed a 24-h IC50 of approximately 27 μM. Due to their low cytotoxicity, we assessed the combined effects of GC-VI-70 and PB4 as antimicrobial agents (see Section 2.2 and Table 3).

#### Evaluation of Antibiotic Cytotoxicity

The MTT assay was also performed for the selected antibiotics to evaluate the effect of their combined treatment with PB4. The results are expressed as a percentage of cell growth compared to the control (results reported in Appendix A), and the IC50 (μM) values were not calculated since only two concentrations were tested, unlike the defined range used for the heteroaryl ethylene compounds. The results are presented in Figure 4 and Table 4. As we were setting up a co-culture model using *E. faecalis* ATCC 29212 and CaCo-2 cells, we chose only two antibiotic concentrations for the cytotoxicity tests: MIC and sub-MIC concentrations reported for *E. faecalis* ATCC 29212 (Table 1). The use of PB4 as an adjuvant to standard antibiotics served two purposes: (i) to control bacterial proliferation by using concentrations of both compounds that did not exert cytotoxic activity on the cells and (ii) to propose a new solution to emerging antibiotic resistance.

After 48 h of treatment, none of the antibiotics altered the physiological cell growth (Table 6), which indeed exceeded T0 and, in some cases, even doubled (ampicillin-MIC 48 h, 189.3%). Regarding the combination of antibiotics and PB4 0.2 μM, no significant difference was noted compared to cells treated with PB4 alone, but the growth was significantly inhibited compared to solutions containing only the antibiotic (mean of 77.7% and 80.7% at 24 and 48 h compared to 87.3% and 158.5%).

To highlight the statistical differences between the treated and untreated cells, a Dunnett’s multiple comparisons test was performed (Appendix A). No remarkable statistical differences emerged for the cells treated with antibiotics alone, with the only exception being observed between the untreated cells and the cells treated with ampicillin at the MIC concentration after 48 h. When cells were treated with antibiotics and PB4, after 24 h, there was a statistical difference only with the cells treated with the gentamicin MIC concentrations, erythromycin sub-MIC concentrations, and rifampicin MIC concentrations. However, these differences were not statistically significant. Nevertheless, after 48 h, the cells treated with antibiotics and PB4 exhibited significant statistical differences compared to the untreated cells, except the rifampicin-treated cells at the MIC and sub-MIC concentrations, and the erythromycin-treated cells at the sub-MIC concentrations.

## 3. Discussion

The threat of multidrug-resistant bacteria poses a grave challenge to global health. Over the years, the misuse and overuse of antibiotics have led to the emergence of bacteria that are resistant to multiple drugs. This resistance significantly reduces the efficacy of standard antibiotic treatments, making once easily treatable infections increasingly difficult to manage. One approach to addressing antimicrobial resistance relies on designing and synthesizing novel molecules with antimicrobial properties, exploring innovative strategies through rational in silico design [24,25]. Leveraging a vast in-house database of heteroaryl ethylene molecules, we developed two QSAR models to select six compounds. Subsequently, we initiated experimental studies to test their antibacterial activity against a range of Gram-positive and Gram-negative bacterial strains. The results reveal promising antimicrobial activity for these molecules, aligning well with the QSAR model for antimicrobial activity against *S. aureus* ATCC 29213. Additionally, we evaluated the molecules in conjunction with commonly used antibiotics to assess any positive effects of combination therapy. Notably, using PB4 alongside various antibiotics targeting protein synthesis, such as rifampin in *S. aureus* and *E. faecalis*, gentamicin against *E. faecalis*, *K. pneumoniae*, and *P. aeruginosa,* significantly reduced the MIC values.

Overall, these results provide essential preliminary data for establishing a eukaryotic–prokaryotic co-culture involving *E. faecalis* ATCC 29212 and CaCo-2 cells. Co-culturing *E. faecalis* with human intestinal epithelial cells has been previously utilized in studies to investigate bacterial adherence, penetration, and host epithelium destruction [26,27]. In the context of this infection model, using molecules like PB4 or GC-VI-70 in combination with antibiotics may allow us to extend the co-culture model while preventing uncontrolled bacterial proliferation and cytotoxic outcomes. The combination of antibiotics, such as gentamicin and rifampin, with PB4 demonstrated significant decreases in MIC values while preserving cellular metabolism. Such combinations will be crucial for setting up co-culture experiments, as all heteroaryl compounds studied here exhibit high cytotoxic activity against colorectal cancer cells.

These results have motivated our research focus on heteroaryl ethylene structures for drug development with multitarget biological activity, particularly against Gram-negative strains [11,28].

The main focus of this multidisciplinary research group is on understanding the mechanisms by which PB4 exhibits antimicrobial and cytotoxic properties.

PB4 shows promise for combination therapies with common antibiotics, enhancing their effectiveness even at very low concentrations, which do not produce cytotoxic effects [29,30,31].

Using PB4 in combination with existing antibiotics could reduce their MIC values, enhancing its potential as a valuable strategy to improve the effectiveness of current antibiotics, reducing antibiotic dosages, and minimizing the risk of side effects and adverse reactions in patients. In addition, combining PB4 with antibiotics that target different aspects of bacterial function may create a synergistic effect, making it harder for bacteria to develop resistance. This approach could lead to a more sustainable and long-lasting solution to combat AMR.

Developing new synthetic molecules with antibacterial activity is of paramount importance, as they offer the potential to overcome existing resistance mechanisms, provide broad-spectrum activity, and reduce toxicity [32]. It is crucial to evaluate the mechanisms of action in terms of their observed combination efficacy. Moreover, conducting in vivo studies will help to assess the safety, pharmacokinetics, and efficacy of these molecules in more complex biological systems. The undeniable importance of future research and investment in discovering and developing such molecules is crucial to counteract the threat of antibiotic resistance and ensure effective treatment options for multidrug-resistant bacterial infections [33,34]. The key point of this work is that the design of new chemical compounds plays a dominant role in medicine [35], enabling the development of innovative drugs with enhanced efficacy and reduced side effects.

## 4. Materials and Methods

### 4.1. Dataset of the QSAR models

Principal component analysis (PCA) and a PLS model were performed using Volsurf+ (VS+) software version 1.1.2, developed by Molecular Discovery, Borehamwood, Hertfordshire, United Kingdom (see Appendix A for further details) [36,37].

The dataset of the QSAR model for antimicrobial activity comprised 53 heteroaromatic compounds (Appendix A) with experimental values for antimicrobial activity against *S. aureus* ATCC29213. The antimicrobial activity was evaluated by means of the standard minimal inhibition concentration (MIC, mg/L).

For the development of the QSAR model for cytotoxic activity, the in-house dataset comprised 38 heteroaryl ethylenes (Appendix A) with experimental in vitro activity values, expressed as logGI50, against the CaCo-2 colon–rectal cancer cell line.

### 4.2. Compound Synthesis

Compounds PB4, BCM4, BCM6, GC-VII-39, GC-VII-50, GC-VI-70, and GC-VI-10 were synthesized via base-catalyzed Knoevenagel condensation between the proper pyridinium/quinolinium ions, as their iodide salts, and the corresponding aldehyde following previously reported procedures [11,12,38,39,40,41].

### 4.3. Bacterial Strains

To assess the antibacterial properties and the spectrum of activity of the six new synthetic chemical compounds, as well as our lead compound, PB4, nine ATCC bacterial strains, both Gram-positive and Gram-negative, were selected (Table 7). For *S. aureus*, two methicillin-susceptible *Staphylococcus aureus* (MSSA) and one MRSA (ATCC BAA-1556) were selected. All of the control strains were provided by the American Type Culture Collection (Manassas, VA, USA). All tests were conducted in duplicate, and both replicates showed the same MIC values.

### 4.4. Bacterial Growth Conditions

The *S. aureus* and *E. faecalis* strains were cultivated respectively on mannitol salt agar (Cat. No. CM0085B) and bile aesculin agar (Cat. No. CM0888), whereas *E. coli*, *A. baumannii*, *P. aeruginosa*, *K. pneumoniae*, and *S. enterica* were cultivated on MacConkey Agar (Cat. No. CM0007). All bacterial strains were incubated overnight at 37 °C. All culture media were purchased from Thermo Scientific^TM^ Oxoid^TM^, Basingstoke, UK.

### 4.5. Antimicrobial Susceptibility Test

The antimicrobial susceptibility test (AST) was conducted by performing broth microdilution (BMD) to determine the minimum inhibitory concentration (MIC) of the selected compounds and PB4, as well as of some well-known antibiotics, such as linezolid (Cat. No. 460592500), gentamicin (Cat. No. 15750037), ampicillin (Cat. No. J60977.06), erythromycin (Cat. No. J62279.09), and rifampin (Cat. No. J60836.03). All antibiotics were provided by Thermo Scientific^TM^ Oxoid^TM^, Basingstoke, UK. The tested concentrations ranged from 128 to 0.125 mg/L.

We employed PB4 to examine its ability to increase the effectiveness of the five antibiotics when used in combination against the ATCC 29212 *E. faecalis* strain. PB4 was used at a fixed concentration of 0.125 μg/mL (0.24 μM), corresponding to a quarter of an MIC value, whereas the antibiotic concentrations ranged from 128 to 0.125 mg/L. BMD was performed using cation-adjusted Müeller Hinton Broth (CA-MHB) (Cat. No. 212322, BD BBL^TM^, Franklin Lakes, NJ, USA) according to standard methods [42]. All antibiotics were solved in an appropriate solvent according to the CLSI guidelines [43]. All heteroaryl ethylene compounds were solved in 100% dimethyl sulfoxide (DMSO) (Cat. No. 85190, Thermo Scientific^TM^ Oxoid^TM^, Basingstoke, UK), obtaining a starting concentration of 8000 mg/L. Intermediate dilutions were carried out using CA-MHB in order to reduce the final DMSO concentration. The highest concentration of tested compounds was 128 mg/L, with a final DMSO concentration in the well of almost 1.6 %. As described in the literature, this DMSO concentration was shown to be well-tolerated by bacteria and it is comparable to the concentration of 1% recommended by the CLSI guidelines [42]. Additionally, we also assessed the effectiveness of the PB4 and GC-VI-70 combination, which are, respectively, a heteroaryl ethylene compound, previously tested in published studies [9], and one of the six tested compounds that showed the best cytotoxicity/MIC profile. The efficacy of the combination was assessed by BMD on Gram-positive ATCC strains: ATCC 29231, ATCC 12598, ATCC BAA-1556 (*S. aureus*), and ATCC 29212 (*E. faecalis*). PB4 was used at the fixed concentration equal to the MIC value for the specific strain, whereas the GC-VI-70 tested concentrations ranged from 128 to 0.125 mg/L.

### 4.6. Evaluation of the Cytotoxic Activity of the Compounds on Human Colorectal Adenocarcinoma Cells

To evaluate the effect of the heteroaryl ethylene compounds and the antibiotics, an MTT ([3-(4,5-dimethylthiazol-2-yl)-2,5-diphenyltetrazolium bromide]) assay was performed as previously described [10]. Briefly, human colorectal adenocarcinoma cells (CaCo-2 HTB-37^TM^, American Type Culture Collection, Manassas, VA, USA) were grown in Dulbecco’s MEM (DMEM) with 10% heat-inactivated fetal bovine serum, 2 mM L-Alanyl-L-Glutamine, penicillin–streptomycin (50 units, 50 μg for mL, just for cells treated with heteroaryl ethylene compounds) and incubated at 37 °C in a humidified atmosphere of 5% CO_2_, 95% air. CaCo-2 cells were plated in 96-well plates and incubated at 37 °C. The heteroaryl ethylene solution and 5-FU (Cat. No. F6627, Merck KGaA, Darmstadt, Germany) were prepared as a 1 mM solution in 10 mL with 0.01% DMSO. In addition to this, the cytotoxic activities of five common antibiotics, linezolid (Cat. No. 460592500), gentamicin (Cat. No. 15750037), ampicillin (Cat. No. J60977.06), erythromycin (Cat. No. J62279.09), and rifampin (Cat. No. J60836.03), were also tested, treating cells at two concentrations corresponding to the MIC and half-MIC values obtained for the ATCC 29212 *E. faecalis* strain. Both concentrations were also tested in combination with PB4 0.2 µM. All antibiotics were provided by Thermo Scientific^TM^ Oxoid^TM^, Basingstoke, UK. A summary of the solutions tested and their relatives’ concentrations are reported in Table 8. Twenty-four hours after plating, the cells were treated with 20 μL of each solution. Untreated cells were used as controls. The microplates were incubated at 37 °C in a humidified atmosphere of 5% CO_2_, 95% air for 24 h, and then the cytotoxicity was measured with a colorimetric assay based on the use of tetrazolium salt MTT (3-(4,5-dimethylthiazol-2-yl)-2,5-diphenyl tetrazolium bromide. IC50: This parameter expresses the concentration of the tested compound necessary to kill half of the cell population after 24 and 48 h of incubation relative to the untreated controls. The absorbance values at 569 nm were obtained using a multi-well plate reader (Synergy H1, Biotek, Via Rodolfo Farneti, 8, 20129 Milano MI). Each value stemmed from an average of four wells. The IC50 values were calculated by nonlinear regression analysis using GraphPad Prism 6.0 software.

## 5. Conclusions

These compounds serve as building blocks for designing novel therapeutic agents [44], targeting previously untreatable diseases. The easy-end versatile synthetic process allows for the optimization of drug properties, such as solubility [45], stability [46], and bioavailability [47], leading to better drug formulations. Additionally, new chemical compounds offer opportunities for discovering new mechanisms of action, expanding the understanding of disease biology and ultimately advancing medical treatment [24]. The application of combination therapy employing novel molecules as adjuvants to existing antibiotics offers several advantages. Firstly, it may restore or enhance the effectiveness of existing antibiotics that have lost their efficacy against resistant strains. Secondly, it can help reduce the overall dosage of antibiotic compounds, potentially minimizing the risk of adverse side effects and decreasing the selective pressure for resistance development. Moreover, the use of effective drug combinations can potentially overcome existing resistance mechanisms, providing useful treatment options against multidrug-resistant bacteria.

## Data Availability

Data supporting reported results will be provided upon request by the corresponding author.

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
