# Peer review of "Evaluation of the Effects of Heteroaryl Ethylene Molecules in Combination with Antibiotics: A Preliminary Study on Control Strains"

_antibiotics, 2023, doi:10.3390/antibiotics12081308_

Round 1

Reviewer 1 Report

The manuscript entitled "Evaluation of synergistic effect of heteroaryl-ethylene molecules in combination with antibiotics: a preliminary study on control strains" assessed the antibacterial properties of the heteroaryl-ethylene molecules in combination with antibiotics against nine ATCC strains (in a synergic assay). This research article explores a fascinating area of study, investigating the potential synergistic effects of heteroaryl-ethylene molecules in combination with antibiotics on control strains. The combination of novel compounds with antibiotics has promising implications for addressing antibiotic resistance and enhancing antimicrobial therapies. The research design and methodology of this preliminary study appear well-thought-out and appropriately structured. The inclusion of control strains helps ensure the validity and reliability of the experimental results. The preliminary findings presented in this article are promising and provide a solid foundation for further investigation. However, some changes can be suggested for further improvement:

Suggestions: 

Discuss the rationale behind the selection/design of heteroaryl-ethylene molecules for this study. 

Perform docking studies to explore the mechanism of these compounds. 

Explain why these specific cells were chosen for the experiment and their relevance to the study's objectives.

Explain the rationale for selecting the five antibiotics (linezolid, gentamycin, ampicillin, erythromycin, rifampin, and imipenem) to test in combination with PB4.

Author Response

Rev1

Comments and Suggestions for Authors

The manuscript entitled "Evaluation of synergistic effect of heteroaryl-ethylene molecules in combination with antibiotics: a preliminary study on control strains" assessed the antibacterial properties of the heteroaryl-ethylene molecules in combination with antibiotics against nine ATCC strains (in a synergic assay). This research article explores a fascinating area of study, investigating the potential synergistic effects of heteroaryl-ethylene molecules in combination with antibiotics on control strains. The combination of novel compounds with antibiotics has promising implications for addressing antibiotic resistance and enhancing antimicrobial therapies. The research design and methodology of this preliminary study appear well-thought-out and appropriately structured. The inclusion of control strains helps ensure the validity and reliability of the experimental results. The preliminary findings presented in this article are promising and provide a solid foundation for further investigation. However, some changes can be suggested for further improvement:

R: We thank the Reviewer for the comments.

Suggestions: 

Discuss the rationale behind the selection/design of heteroaryl-ethylene molecules for this study. 

R: Section 2.1 (lines 93-108) describe the selection/design of the new drug candidates. We have extended this section according to Reviewer suggestion.

Perform docking studies to explore the mechanism of these compounds. 

R: This kind of study implies the knowledge of the mechanism of action and a precise molecular target (i.e. . Unfortnately, the use of heteroaryl-ethylene molecules is only at the early pioneering phases and no information are still available. In the future (and in some preliminary experiment) we have study a preliminary mechanism of action with mRNA seq compared to 5-Fu on Caco2 cells. We using a custom panel comprising 56 genes, involved in Oxidative stress, immunoregulation and inflammation, apoptosis and other pathways involved in different cellular process such as: metalloproteinase, nAChr, ATP Binding Protein. PB4 shows a dramatic gene silencing impact on most genes, probably depending on its acute toxicity, except for TXN2 and CHRNA2 (Oxidative stress and nAChr). GC-VI-70 instead has a similar trend to 5-FU, reporting a variation of gene expression analogous to 5-FU, especially for TGFB1, IL8 (immunoregulation and inflammation), NOX1, KEAP1, CAT (Oxidative stress) and ABCC1 (ATP Binding Cassette). These results encourage further investigation to match new compounds synthesis and the molecular pathway they impact. CG-VI-70 show an apparent 5FU mimicry; this behavior and the modulation of the cellular metabolism make it an interesting candidate for further studies for pharmaceutical purposes to take action on alterated biochemical pathway. We hope to obtained the results for a new paper dedicated to this part in early time.

Explain why these specific cells were chosen for the experiment and their relevance to the study's objectives.

R: CaCo2 cells are a human colon epithelial cancer cell line that is used as a model of human intestinal drug and compound absorption that are commonly used to simulate the intestinal epithelial barrier. CaCo2 cells, which have already been used in early experiments with PB4 (PMID: 34572616), differentiate to form tight junctions between cells when cultured as a monolayer, serving as a model of paracellular compound movement across the monolayer. CaCo2 cells also express transporter proteins, efflux proteins, and Phase II conjugation enzymes in order to model a variety of transcellular pathways as well as the metabolic transformation of test substances. The CaCo2 cell monolayer closely resembles the human intestinal epithelium in many ways (PMID: 16922635, PMID: 29787057). Our research group then has a great previous experience in using this specific cell line, with various data established over time, including eukaryotic/ procaryotic co-culture and various tests with substances. (https://doi.org/10.1021/acs.jnatprod.6b00577, DOI: 10.1021/acs.jnatprod.5b00619, https://doi.org/10.1039/C3OB42521E). For all these reasons, this specific cell line was chosen.

Explain the rationale for selecting the five antibiotics (linezolid, gentamycin, ampicillin, erythromycin, rifampin, and imipenem) to test in combination with PB4.

R: All antibiotics tested, were chosen according to their class, so as to test the widest possible spectrum, also according to the different bacterial species, in combination with PB4. The choice of these antibiotics was also dictated by our previous knowledge of their impact on cell viability, so as to have a clear view of the use in combination with the PB4 molecule.

Reviewer 2 Report

The manuscript titled "Evaluation of synergistic effect of heteroaryl-ethylene compounds in combination with antibiotics: a preliminary study on control strains" addresses the urgent concern of multidrug-resistant bacteria and explores the potential of using heteroaryl-ethylene compounds in combination with antibiotics to overcome this issue. The study focuses on investigating the cytotoxic and antimicrobial activities of these compounds and their efficacy when used in combination therapy.

In evaluating the cytotoxicity of these compounds, the researchers employed the MTT assay on human colorectal adenocarcinoma cells (CaCo-2). The results revealed significant cytotoxic activity, surpassing that of the reference compound 5-Fluorouracil. The compounds BCM6, GC-VII-39, and GC-VII-50 demonstrated particularly high activity after 24 hours. The study also explored the potential of combining these compounds with antibiotics, and it was found that certain combinations exhibited synergistic effects, with the addition of compound PB4 significantly reducing the minimum inhibitory concentration (MIC) of antibiotics against specific bacterial strains.

The paper highlights the importance of using heteroaryl-ethylene compounds as building blocks for designing new therapeutic agents. Their versatility allows for optimization of drug properties, potentially leading to improved drug formulations. The study emphasizes the advantages of combination therapy, which can enhance antibiotic efficacy against resistant strains and reduce the overall antibiotic dosage. This strategy also has the potential to minimize adverse side effects and mitigate the development of resistance.

In conclusion, this paper provides a comprehensive evaluation of the cytotoxic and antimicrobial activities of heteroaryl-ethylene compounds and their potential in combination therapy with antibiotics. It highlights the promising synergistic effects observed and demonstrates the potential of these compounds as effective treatment options against multidrug-resistant bacteria. Further research and development in this field are crucial for achieving effective therapeutic strategies for combating antibiotic resistance and improving medical treatment options.

Author Response

Rev2

Comments and Suggestions for Authors

The manuscript titled "Evaluation of synergistic effect of heteroaryl-ethylene compounds in combination with antibiotics: a preliminary study on control strains" addresses the urgent concern of multidrug-resistant bacteria and explores the potential of using heteroaryl-ethylene compounds in combination with antibiotics to overcome this issue. The study focuses on investigating the cytotoxic and antimicrobial activities of these compounds and their efficacy when used in combination therapy.

In evaluating the cytotoxicity of these compounds, the researchers employed the MTT assay on human colorectal adenocarcinoma cells (CaCo-2). The results revealed significant cytotoxic activity, surpassing that of the reference compound 5-Fluorouracil. The compounds BCM6, GC-VII-39, and GC-VII-50 demonstrated particularly high activity after 24 hours. The study also explored the potential of combining these compounds with antibiotics, and it was found that certain combinations exhibited synergistic effects, with the addition of compound PB4 significantly reducing the minimum inhibitory concentration (MIC) of antibiotics against specific bacterial strains.

The paper highlights the importance of using heteroaryl-ethylene compounds as building blocks for designing new therapeutic agents. Their versatility allows for optimization of drug properties, potentially leading to improved drug formulations. The study emphasizes the advantages of combination therapy, which can enhance antibiotic efficacy against resistant strains and reduce the overall antibiotic dosage. This strategy also has the potential to minimize adverse side effects and mitigate the development of resistance.

In conclusion, this paper provides a comprehensive evaluation of the cytotoxic and antimicrobial activities of heteroaryl-ethylene compounds and their potential in combination therapy with antibiotics. It highlights the promising synergistic effects observed and demonstrates the potential of these compounds as effective treatment options against multidrug-resistant bacteria. Further research and development in this field are crucial for achieving effective therapeutic strategies for combating antibiotic resistance and improving medical treatment options.

R: We thank the Reviewer for the comments. Many thanks for our groups is a great pleasure receive this comments, we hope to increment this type of knowledge with other work.

Reviewer 3 Report

1. Earlier publications of the author with the same title compound projected as a anti-microbial ability with great efficacy. 

2. It would be grateful if the author explains the mode of antibacterial function of the title compound towards different bacterial strains. 

3.The author explained cell line study, it would be nice if we could show the cell line result images for readers.

4. Though out the article commas (,) have to be replaced by dot (.) separation while representing micromolar concentration.

5. Overall, the article and the scientific content is very essential for scientific community to address. 

Author Response

Rev3

Comments and Suggestions for Authors

  1. Earlier publications of the author with the same title compound projected as a anti-microbial ability with great efficacy. 

R: We thank the Reviewer for the comments. Many thanks for our groups is a great pleasure receive this comments, we hope to increment this type of knowledge with other work.

  1. It would be grateful if the author explains the mode of antibacterial function of the title compound towards different bacterial strains. 

R: Many thanks the use of heteroaryl-ethylene molecules is only at the early pioneering phases and no information are still available. We are just working up some background information to shed light on the mechanism of action.

3.The author explained cell line study, it would be nice if we could show the cell line result images for readers.

R: Cell images have not been reported, as they do not explain very well the impact of substances on cell viability, in our opinion, often the cells are similar to live cells but in a “off metabolic status”, and appear how a point. The MTT viability test provides all the necessary information in this regard. Another set of experiment will be dedicated to identify localization of  cellular compartment and dosage of free-radicals and other biochemical line.

  1. Though out the article commas (,) have to be replaced by dot (.) separation while representing micromolar concentration.

R: We have corrected the typos according to reviewer suggestion.

  1. Overall, the article and the scientific content is very essential for scientific community to address. 
    R: We thank the Reviewer for the comments.

Reviewer 4 Report

The article entitled “Evaluation of synergistic effect of heteroaryl-ethylene molecules in combination with antibiotics: a preliminary study on control strains” was reported by Carmelo Bonomo and his group. A major revision is suggested based on the following comments.

Comments:

[1] Both the structure in scheme 2 and the graphs in figure 2 are blurry.

[2] In the script, there should be a space between the text and the reference.

[3] All references in the running text should appear first, followed by a period or a comma. Certain references are marked in italics and bold font.

[4] Some paragraphs in the script are improperly aligned.

[5] In Table 3, values and text are combined.

Author Response

Rev4

Comments and Suggestions for Authors The article entitled “Evaluation of synergistic effect of heteroaryl-ethylene molecules in combination with antibiotics: a preliminary study on control strains” was reported by Carmelo Bonomo and his group. A major revision is suggested based on the following comments.

[1] Both the structure in scheme 2 and the graphs in figure 2 are blurry. 
R: The quality of both scheme and picture has been improved according to reviewer comment.

[2] In the script, there should be a space between the text and the reference.

R: We have corrected the typos according to reviewer suggestion.

[3] All references in the running text should appear first, followed by a period or a comma. Certain references are marked in italics and bold font.

R: We have corrected the typos according to reviewer suggestion.

 [4] Some paragraphs in the script are improperly aligned.

R: We have corrected the typos according to reviewer suggestion.

[5] In Table 3, values and text are combined.

R: We have corrected the typos according to reviewer suggestion.

Reviewer 5 Report

Tha study titled “Evaluation of synergistic effect of heteroaryl-ethylene molecules in combination with antibiotics: a preliminary study on control strains.” Is an interesting study but there are several drawbacks in the study. The authors have failed to maintain controls and provide statistical studies to prove the significance of the values. I have provided few comments that the author should address to improve the quality of the manuscript.

1.      The authors should consider moving 2.1 to methodology. Since the compounds were not synthesized in the study, this should be removed from the results.

2.      There is unnecessary capitalization of words throughout the manuscript. This should be corrected.

3.      The structure of PB4 is missing in the manuscript. The synthesis information of PB4 is not mentioned anywhere nor any reference is provided. This should be shown in scheme 2

4.      The molecular weights of the compounds should be given.

5.      Figure 1and 2: What does t(1), t(2) and t(3) mean in the axis labels. If it is the latent variables then it should be renamed appropriately.

6.      Figure 1 and 2 : What do the grey circles in figure 1 and 2 mean? This should be given in the figure caption.

7.      Line 137: unionized species? Do the authors mean non-ionized? Correct the same.

8.      The representation of CaCo2 cells should be uniform. It should either be CaCo2 or CACO-2.

9.      Line 147: There is no inset in figure 1.

10.   Line 206: “compounds” whereas there is just one compound.

11.   All bacterial names should be represented in italics.

12.   The authors should either maintain the manuscript in present tense or paste tense. The language of the manuscript should be improved.

13.   Table 1: The authors should add a positive control and add to the table

14.   Table 1: Standard error of values are missing. The sem values should be mentioned in the table.

15.   Table 1: What does * mean in the table?

16.   Table 1: The authors have presented the most promising MIC values in bold but the values of PB4 are far better than the bold one. Why are the other values bolded in that case?

17.   The concern should be addressed by the authors. Usually, to prove the synergism, antagonism or neutral activity of the compounds, the authors should perform fractional inhibitory concentration index experiments. Refer to https://doi.org/10.1128%2FAAC.00999-09 , https://doi.org/10.1093/jac/11.5.427  and https://doi.org/10.1111/febs.16770 . The experiments conducted by the authors do not prove that there is synergism between the compounds. The FIC experiment should be performed mandatorily to prove this.

18.   Table 2 caption: “most significant decrease” what is the significance calculated too? What method was used? What was the p value for the significance?

19.   Address for table 3 with respect to comment above.

20.   Figure 1: dose response curve: How is this figure 1 again?

21.    Figure 1: dose response curve: (A) and (B) How can the percentage cell growth be negative? There are several points for the samples that have negative percentage cell growth. This is not possible.

22.   Table 4: Show the table as column graph and prove statistical significance. The authors should also include the sem value and a positive control.

23.   Line 312: PB4 as adjuvant? The word adjuvant means differently and an adjuvant is usually used to elicit a stronger immune response. What do the authors mean here?

24.   The authors have used “,” instead of dots “.” at all instances of decimal point. This needs to be corrected as use of “,” instead of a decimal point creates confusion.

25.   Table 5: Make corrections as with respect to comment made above.

26.   It is not clear as to what the authors mean by co-culture. Did the authors culture the bacteria and the CaCo2 cells together in a cell flask? If this is true, how did the authors measure the MTT directly without measuring the dead bacterial cells? MTT stain both bacterial and CaCo2 cells. So the value presented is a cumulative of both CaCo2 and bacteria. The authors have not mentioned the methods of this section in the methodology and also, the question should be answered in the manuscript too.

27.   Discussion is very vague and needs to be more focussed.

The manuscript required extensive proof-reading. 

Author Response

Rev5:

Comments and Suggestions for Authors
Tha study titled “Evaluation of synergistic effect of heteroaryl-ethylene molecules in combination with antibiotics: a preliminary study on control strains.” Is an interesting study but there are several drawbacks in the study. The authors have failed to maintain controls and provide statistical studies to prove the significance of the values. I have provided few comments that the author should address to improve the quality of the manuscript.

  1. The authors should consider moving 2.1 to methodology. Since the compounds were not synthesized in the study, this should be removed from the results.

R: Section 2.1 discuss the rationale behind the selection/design of the heteroaryl-ethylene molecules employed for this study, this is why the authors value and believe crucial this section. The authors want to reassure the reviewer that, for any new study, new batch are prepared. Nevertheless, the synthetic procedures have already been reported in previous studies and we have modified the title of this section according to reviewer comment.

  1. There is unnecessary capitalization of words throughout the manuscript. This should be corrected.

R: We have corrected the typos according to reviewer suggestion.

  1. The structure of PB4 is missing in the manuscript. The synthesis information of PB4 is not mentioned anywhere nor any reference is provided. This should be shown in scheme 2

R: Maybe the Reviewer is inadvertently missing something. The structure and useful information about PB4 have been already inserted by the authors in the introduction section, lines 52-57 (Scheme 1 and reference 9-10), and Scheme 2 refers to the six heteroaryl ethylenes employed in this study. The information on the synthesis of all the compounds have already been indicated in section 4.2 and proper reference have already been provided.

  1. The molecular weights of the compounds should be given.

R: Scheme 2 has been improved according to reviewer comment.

  1. Figure 1and 2: What does t(1), t(2) and t(3) mean in the axis labels. If it is the latent variables then it should be renamed appropriately.

R: PLS works decomposing the X-matrix as the product of two smaller matrices:
-The loading matrix (P) contains information about the variables. It contains a few vectors (Latent Variables, LVs) which are linear combinations of the original X-variables. The concept of LV is quite equivalent to the PC in PCA.
-The score matrix (T) contains information about the objects.
Each object is described in terms of the t scores for each LVs, for these reasons the two terms are often used interchangeably.
We thank the Reviewer for the comment and we have modified the captions of both Figure 1 and 2 according to this comment.

  1. Figure 1 and 2 : What do the grey circles in figure 1 and 2 mean? This should be given in the figure caption.

R: The two grey ellipsoids represent the T-hotelling plot (95% confidence interval for the outer surface and 99% confidence interval for the inner surface). We have modified the captions of both Figure 1 and 2 according to this comment.

  1. Line 137: unionized species? Do the authors mean non-ionized? Correct the same.

R: The Volsurf+ software uses different molecular descriptors related to charge state. The % unionised species (%FU4 - %FU10) represents the percentage of unionised species is calculated at pH 4, 5, 6, 7, 8, 9 and 10. Obviously, ‘un-‘ is the prefix meaning ‘not’, while ‘ionise’ is the British variant of the verb ‘ionize’, but this is the software ‘code’ and we cannot correct it.

  1. The representation of CaCo2 cells should be uniform. It should either be CaCo2 or CACO-2.

R: We have corrected the typos according to reviewer suggestion.

  1. Line 147: There is no inset in figure 1.

R: We have corrected the typos according to reviewer suggestion Now line 155

  1. Line 206: “compounds” whereas there is just one compound.

R: We have corrected the typos according to reviewer suggestion.

  1. All bacterial names should be represented in italics.

R: We have corrected the typos according to reviewer suggestion.

  1. The authors should either maintain the manuscript in present tense or paste tense. The language of the manuscript should be improved.

R: Dear reviewer all text are revised for the English language

  1. Table 1: The authors should add a positive control and add to the table

R: We added Table 2 showing the MIC values for the Quality Control strains. We chose gentamicin and the strains were selected following the EUCAST QC guidelines (E. coli ATCC 25922 for Enterobacterales; P. aeruginosa ATCC 27853 for P. aeruginosa and A. baumannii; S. aureus ATCC 29213 for S. aureus; E. faecalis ATCC 29212 for E. faecalis).

  1. Table 1: Standard error of values are missing. The sem values should be mentioned in the table.

R: As mentioned in the Methodology section (Lines 592-593), both the replicates showed exactly the same MIC values with no variability. We added a statement in the Results section (LINES 311-313).

  1. Table 1: What does * mean in the table?

R: We have corrected the typos according to reviewer suggestion.

  1. Table 1: The authors have presented the most promising MIC values in bold but the values of PB4 are far better than the bold one. Why are the other values bolded in that case?

R: We bolded only the MIC values referred to the new compounds tested in this study, whereas PB4 has been already tested and published (see ref 9).

  1. The concern should be addressed by the authors. Usually, to prove the synergism, antagonism or neutral activity of the compounds, the authors should perform fractional inhibitory concentration index experiments. Refer to https://doi.org/10.1128%2FAAC.00999-09 , https://doi.org/10.1093/jac/11.5.427  and https://doi.org/10.1111/febs.16770 . The experiments conducted by the authors do not prove that there is synergism between the compounds. The FIC experiment should be performed mandatorily to prove this.

R: Thanks the reviewer for the comment. We made a lexical mistake. What we wanted to highlight with “synergism” is the ability of a compound to induce the decrease of MIC values of another one. Therefore, we rephrased the sentences along the text according to the comment.

  1. Table 2 caption: “most significant decrease” what is the significance calculated too? What method was used? What was the p value for the significance?

R: Thanks, for the tip. We substituted the term “significant” with “notable”, “relevant” or “remarkable” which are more appropriate to real meaning of those sentences. We fixed the issue along the entire text.

  1. Address for table 3 with respect to comment above.

R: See the answer above

  1. Figure 1: dose response curve: How is this figure 1 again?

R: We have corrected the typos according to reviewer suggestion.

  1. Figure 1: dose response curve: (A) and (B) How can the percentage cell growth be negative? There are several points for the samples that have negative percentage cell growth. This is not possible.

R: The curves corresponding to cell growth in graph one never fall below the x-axis corresponding to zero. The points to which you refer are the standard deviations calculated by the software, but of course it would not be possible to really have negative values.

  1. Table 4: Show the table as column graph and prove statistical significance. The authors should also include the sem value and a positive control.

R: In our work, as well as in others previously published both by our research group (PMID: 34572616, 25131935) and other groups (PMID: 31338398, 33408116) the value of IC50 is reported as a sigmoid curve and not as a histogram for specific reasons.

First of all, the substances presented in the table were tested at different concentrations (from 100 μM to 0.01 μM) quadrupled on the CaCo-2 cell line, performing an MTT assay at 24 and 48 hours.

Once the values were obtained, the average of each quadrupling group and its standard deviation were calculated. Subsequently, both the arithmetic means and the standard deviations obtained from the four values were normalized as required by the guidelines (PMID: 18582601, 22328315) with respect to T0 following equation (1) for the arithmetic mean and equation (2) for the standard deviation:

Once this normalization was carried out, the values for each concentration and standard deviations were reported on GraphPad Prism together with the relative concentrations tested, and a "log(inhibitor) vs. normalized response" analysis was performed, to determine the IC50 of the compound, the concentration that provokes a response equal to 50%.

Questo tipo di analisi, infatti, prevede che I dati siano stati precedentemente normalizzati (https://www.graphpad.com/guides/prism/latest/curve-fitting/reg_dr_inhibit_normalized.htm). This model assumes that the dose response curve has a standard slope, equal to a Hill slope (or slope factor) of -1.0; this is preferable when there are not many replicates (n > 50).

The calculated model follows the equation (3)

In equation (3) y dependent variable corresponds in our analysis to the percentage of metabolically active cells, while x corresponds to the logarithm of the concentration of the substance (See Figure A). Plotting a straight line for y = 50 yields by projection the value of IC50.

Figure A. Standard Sigmoidal Curves for a “log(inhibitor) vs. normalized response” analysis.

For all these reasons, the graph cannot be represented as a histogram; the standard deviation as he could see is normalized and included in the analysis performed by GraphPad (The points of 95% CI are included in the sigmoids) and the positive control in this analysis is identified rather by the controls, which do not appear in the graphs as they are used for normalization.

  1. Line 312: PB4 as adjuvant? The word adjuvant means differently and an adjuvant is usually used to elicit a stronger immune response. What do the authors mean here?

R:   Dear reviewer, the word “adjuvant” was used in this case with the meaning of “adjuvant therapy”.The idea comes from, for example, chemotherapy adjuvants and multicomponent anti-infective strategies (doi: https://doi.org/10.1136/bmj.321.7270.1208, DOI:https://doi.org/10.1093/annonc/mdl029, DOI: https://doi.org/10.1017/S1462399410001766, DOI:https://doi.org/10.1016/j.tim.2016.06.009). So we mean Adjuvant as: “helpful, assisting, auxiliary”( from Latin ad to, and juvare to help), intending the PB4 as an aid beside the conventional antibiotic therapy.

  1. The authors have used “,” instead of dots “.” at all instances of decimal point. This needs to be corrected as use of “,” instead of a decimal point creates confusion.

R: We have corrected the typos according to reviewer suggestion.

  1. Table 5: Make corrections as with respect to comment made above

R: In the evaluation of the cytotoxicity of antibiotics, we decided not to consider IC50 as a study parameter, but rather the percentage cell growth compared to the control for two essential reasons:

  1. Antibiotics are substances known not to be particularly toxic to cells (especially cancer cells).
  2. only two concentrations were tested, and not enough concentrations to conduct a satisfactory IC50 test.

Consequently, for each antibiotic two concentrations were tested as explained in the work: the concentration of MIC and a concentration equal to MIC/2. Both were tested alone and in combination with PB4 0.2 μM at 24 and 48 hours. Subsequently, a MTT assay was performed and values such as those shown in Figure B were obtained.

Figure B. Matrix obtained from the MTT assay. The boxes in blue contain the treaties, while the first four lines (in green at 24 hours and in yellow at 48 hours) contain the controls.

All four plates were subsequently normalized with respect to T0, especially following equation (4).

Then, each value obtained from the MTT assay was multiplied by 100 and divided by the arithmetic mean of the absorbance values of T0. In this case, therefore, what is obtained is a percentage of growth compared to the initial condition represented by T0.

It is important to note that the absorbance values related to the control have also been normalized according to (4) and this is to be considered as positive control.

Once all the normalized values were obtained, they were processed with GraphPad Prism and the histograms presented before this table were obtained. Within each Histogram it is possible to observe the replicates for each condition (4 replicated for each treatment, 40 replicated for controls and 8 replicated for T0) in addition to the relative confidence interval bars.

In Table S3 the results of a Dunnett's Test (also called Dunnett's Method or Dunnett's Multiple Comparison) are reported; this test compares means from several experimental groups against a control group mean to see is there is a difference.

On the advice of the Reviewer, we insert a further table in the supplementary material containing the normalized values and their standard deviations (Table S4).

  1. It is not clear as to what the authors mean by co-culture. Did the authors culture the bacteria and the CaCo2 cells together in a cell flask? If this is true, how did the authors measure the MTT directly without measuring the dead bacterial cells? MTT stain both bacterial and CaCo2 cells. So the value presented is a cumulative of both CaCo2 and bacteria. The authors have not mentioned the methods of this section in the methodology and also, the question should be answered in the manuscript too.

R:  By co-culture we mean the development of an in vitro experimental model to study the host pathogen interaction. In this article, we mention this in future perspective, as it would be interesting to test the combinations we tested in an in vitro infection context, because this work could be part of a larger project of co-culture experiments planned in the future. All the MTT tests reported in this study were not performed on co-cultures but on the cell line alone, in order to determine the impact of the aforementioned combination on cell viability. In any case, due to our previous experience, we have already addressed the problem you raised . In a previous experiment (DOI: 10.3390/biom11010072) we carried out an additional MTT control-experiment in which five different concentrations of bacteria were analyzed under the same condition employed for cells. The absorbance values at 569 nm were virtually undetectable at the lowest bacterial concentration that went up to 0.03 (still very low) with ∼3300 bacteria. Since for our experiments we used a concentration of bacteria ranging between these two concentrations points, we believe that the presence of bacteria did not significantly influenced the MTT results. It is also worth noting that the values measured at both concentrations were obtained for “free” bacteria, a very different condition compared to the intracellular hostile environment, where the ability of bacteria to metabolize the MTT salt should be even lower. A very different outcome was observed when measuring the absorbance values at 569 nm coming from ∼21000 (average of 0.146) and ∼85000 (average of 0.465) bacteria/well; in case of future experiments employing this concentrations, the absorbance values coming from the presence of bacteria should be taken into account.

  1. Discussion is very vague and needs to be more focussed.

R:  Dear reviewer we have worked on the discussion how your comments and suggestion. We working on:

Line 1185-1190

Line: 1196-1200

Line: 1202-1205

Line: 1384-1394

Round 2

Reviewer 4 Report

Dear Editor,

There are no more comments for this work in revision. I recommend that the current version of the manuscript be accepted for publication in this journal.

Reviewer 5 Report

The authors have addressed all the comments satisfactorily.